# Role of Radiology in the Preoperative Detection of Arterial Calcification and Celiac Trunk Stenosis and Its Association with Anastomotic Leakage Post Esophagectomy, an Up-to-Date Review of the Literature

**DOI:** 10.3390/cancers14041016

**Published:** 2022-02-17

**Authors:** Antonios Tzortzakakis, Georgios Kalarakis, Biying Huang, Eleni Terezaki, Emmanouil Koltsakis, Aristotelis Kechagias, Andrianos Tsekrekos, Ioannis Rouvelas

**Affiliations:** 1Department of Clinical Science, Division of Radiology, Intervention and Technology (CLINTEC), Karolinska Institutet, 141 86 Stockholm, Sweden; antonios.tzortzakakis@ki.se (A.T.); georgios.kalarakis@ki.se (G.K.); 2Medical Radiation Physics and Nuclear Medicine, Functional Unit of Nuclear Medicine, Karolinska University Hospital, 141 86 Huddinge, Sweden; 3Department of Radiology, Karolinska University Hospital, Huddinge, 141 86 Stockholm, Sweden; 4Department of Upper Abdominal Surgery, Karolinska University Hospital Huddinge, 141 86 Stockholm, Sweden; biying.huang93@gmail.com (B.H.); andrianos.tsekrekos@regionstockholm.se (A.T.); 5Department of General Surgery, Södertälje Hospital, 152 86 Södertälje, Sweden; 6Department of Emergency Medicine, Karolinska University Hospital, 171 64 Stockholm, Sweden; terezakielena@gmail.com; 7Department of Radiology, Karolinska University Hospital, Solna, 171 64 Stockholm, Sweden; emmanouilkolts@gmail.com; 8Department of Digestive Surgery, Kanta-Häme Central Hospital, 13530 Hämeenlinna, Finland; aristotelis.kechagias@yahoo.gr; 9Department of Clinical Science, Division of Surgery, Intervention and Technology (CLINTEC), Karolinska Institutet, 141 86 Stockholm, Sweden

**Keywords:** esophageal cancer, esophagectomy, anastomotic leak, risk factors, computed tomography, arterial calcification, celiac trunk stenosis

## Abstract

**Simple Summary:**

Esophageal cancer is the sixth deadliest among all cancers worldwide. Multimodal treatment, including surgical resection of the esophagus, offers the potential for cure even in advanced cases, but esophagectomy is still associated with serious complications. Among these, anastomotic leakage has the most significant clinical impact, both in terms of prognosis and health-related quality of life. Identifying patients at a high risk for leakage is of great importance in order to modify their treatment and, if possible, avoid this complication. This review aims to study the current literature regarding the role of radiology in detecting potential risk factors associated with anastomotic leakage. The measurement of calcium plaques on the aorta, as well as the detection of narrowing of the celiac trunk and its branches, can be easily assessed by preoperative computed tomography, and can be used to individualize perioperative patient management to effectively reduce the rate of leakage.

**Abstract:**

Surgical resection of the esophagus remains a critical component of the multimodal treatment of esophageal cancer. Anastomotic leakage (AL) is the most significant complication following esophagectomy, in terms of clinical implications. Identifying risk factors for AL is important for modifying patient management and improving surgical outcomes. This review aims to examine the role of radiological risk factors for AL after esophagectomy, and in particular, arterial calcification and celiac trunk stenosis. Eligible publications prior to 25 August 2021 were retrieved from Medline and Google Scholar using a predefined search algorithm. A total of 68 publications were identified, of which 9 original studies remained for in-depth analysis. The majority of these studies found correlations between calcifications in the aorta, celiac trunk, and right post-celiac arteries and AL following esophagectomy. Some studies suggest celiac trunk stenosis as a more appropriate surrogate. Our up-to-date review highlights the need for automated quantification of aortic calcifications, as well as the degree of celiac trunk stenosis in preoperative computed tomography in patients undergoing esophagectomy, to obtain robust and reproducible measurements that can be used for a definite correlation.

## 1. Introduction

Despite continuous diagnostic and therapeutic advancements, esophageal and gastroesophageal junction carcinoma remains a major cause of cancer-related mortality worldwide [1]. Esophageal cancer ranks as the seventh most common form of cancer and sixth in regard to mortality, having a poor five-year survival rate of just 20% [2]. Among the available management options, multimodal treatments, combining esophageal resection with neoadjuvant chemoradiation or perioperative chemotherapy offers the best chance of cure for patients with non-metastatic esophageal cancer [3,4,5]. Transthoracic esophagectomy with gastric conduit (GC) reconstruction has been the standard surgical technique used for the treatment of esophageal cancer in specialized gastroesophageal oncologic centers [3]. Depending on tumor location, esophagectomy is performed by either the McKeown procedure with a cervical anastomosis, or the Ivor Lewis procedure with an intrathoracic anastomosis [6,7,8]. Another surgical approach, namely transhiatal esophagectomy with cervical anastomosis, is usually reserved for patients with distally located or junctional tumors and higher comorbidity, as it is less invasive, but this approach is associated with inferior oncological outcomes [9]. In recent years, various minimally invasive techniques have gained popularity as studies have demonstrated at least equal short-term postoperative outcomes and improved long-term survival [10,11,12].

Anastomotic leakage (AL) is one of the most dreaded complications following esophagectomy Figure 1. While the overall 30-day mortality in patients undergoing esophagectomy is 2–3%, it can increase to 17–35% in patients with a sustained leakage [13]. In addition to increased mortality, AL is related to significant postoperative morbidity, prolonged hospital stays, increased recurrence rates, and a worse long-term quality of life [14,15]. The overall incidence of AL after esophagectomy ranges between 10 and 20%, with an associated mortality of 5–10%, indicating a wide variety in the surgical practices used in esophageal cancer and the management of AL worldwide [16]. A consensus with regard to the definition of an AL is still under discussion, and it is a continuous challenge for the scientific community to understand, prevent, diagnose, and treat AL which is a critical, costly, and potentially lethal postoperative complication [17].

In an effort to standardize the diagnosis and management of complications after esophagectomy, the Esophageal Complications Consensus Group (ECCG) updated the definition and classification of AL in 2015 [18]. According to the ECCG, an AL is defined as a ‘full-thickness defect involving the esophagus, anastomosis, staple line or conduit’ [19]. AL is classified into three types with increasing severity. Type 1 AL is treated medically or with dietary modification, and Type 2 AL requires radiologic or endoscopic intervention but not surgical therapy, while Type 3 AL needs surgical intervention [18]. Laboratory tests, radiological imaging, or endoscopy can be used to detect AL [20]. Computed tomography (CT) of the thorax with oral and intravenous contrast provides a high sensitivity and specificity for the detection of AL, and serves nowadays as the diagnostic method of choice. Endoscopic assessment of the upper gastrointestinal tract is used as a second-hand method to confirm the diagnosis and treat AL in severe cases [21]. GC necrosis is another postoperative, though less common, complication after esophagectomy. Type 1 and 2 conduit necrosis (focal), are not associated with AL, while Type 3 conduit necrosis represents a totally necrotic GC accompanied by AL, requiring conduit resection and esophageal diversion [18,21].

Several risk factors for AL have been identified, related to either the patient or the procedure. Among these, congestive heart failure, hypertension, renal insufficiency, diabetes, smoking, previous neoadjuvant therapy, duration of the operation, and a need for blood transfusion are the most common [22]. All these factors may contribute to the ischemia of the reconstructed GC, which ultimately leads to AL [23]. The rate of AL seems to be the highest after transhiatal esophagectomy, followed by McKeown esophagectomy, while Ivor Lewis esophagectomy has the lowest risk of AL [24]. The most cranial part of the GC used for the construction of the anastomosis is particularly vulnerable in terms of arterial supply, since it is exclusively supplied by the right gastroepiploic artery, one of the terminal branches of the gastroduodenal artery deriving from the common hepatic artery [25]. The anatomy of the gastroduodenal artery varies though, and this vessel may also arise from the left or the right hepatic arteries, or even directly from the celiac trunk [26].

According to European guidelines, a CT or ^18^F-FDG PET/CT scan of the neck, thorax, and abdomen should be performed for the staging of esophageal cancer prior to treatment initiation [27]. Since, routinely, a full-body CT scan is available for each patient undergoing esophagectomy, the evaluation of radiological risk factors predicting poor outcomes, and in particular AL, can be easily incorporated into everyday practice Figure 2. A recent review by Knight et al. [28] and a meta-analysis by Hoek et al. [29] examined the association between AL and arterial calcification following esophagectomy. Both studies concluded that there is significant evidence pointing to the contribution of calcification in the thoracic aorta, the celiac axis, and the right post-celiac arteries in the development of postoperative AL for patients undergoing esophagectomy with GC reconstruction. Other studies have demonstrated a correlation between the degree of celiac trunk stenosis on preoperative CT and AL [30,31].

This up-to-date literature review aims to examine the preoperative radiological assessment of aortic calcification and celiac trunk stenosis as risk factors for AL after esophagectomy, by summarizing all of the relevant published original studies, and explore its implications in clinical practice.

## 2. Methods

The report of our results was performed according to PRISMA (preferred reporting items for systematic reviews and meta-analysis) guidelines (register number: 1298) [32].

### 2.1. Search Strategies

A literature search was conducted to identify relevant publications in Medline and Google Scholar. The following search algorithm was used: [(aortic calcification OR celiac trunk stenosis) AND (esophagectomy) AND (anastomotic leakage OR esophageal conduit necrosis)]. The search was restricted to studies in human subjects and published in English in full text prior to 25 August 2021. The reference lists of the selected articles were also screened in order to identify additional publications of relevance.

### 2.2. Study Selection and Data Extraction

Studies were considered eligible if they fulfilled the following criteria: original articles reporting on either the examination of aortic calcification or celiac trunk stenosis as possible risk factors for AL after esophagectomy with GC reconstruction. The exclusion criteria were as follows: (i) reviews, meta-analyses, or case reports, (ii) duplicates or studies reporting on the same patient cohort, (iii) studies where esophagectomy of a type other than GC reconstruction was performed.

All eligible publications were independently screened for their relevance by 3 reviewers (E.T., G.K., and A.T.). The same reviewers extracted relevant data: study characteristics (first author, country, year of publication, study type, number of patients, and time period of inclusion), type of surgery (McKeown, Ivor Lewis, other), type of exposure/radiological correlation (aortic calcification, celiac trunk stenosis), and AL rate post esophagectomy. Discrepancies in findings were discussed among the 3 reviewers and subsequently resolved.

## 3. Results

A total of 68 potentially relevant articles were identified through an electronic search of bibliographic databases and a manual search of reference lists, of which 34 were directly excluded as duplicates. Of the remaining 34 articles that underwent full-text evaluation, 25 were excluded for the following reasons: 1 case report of 2 patients [33], 1 systematic review [28], 1 systematic review and meta-analysis [29], and 22 articles which were assessed as irrelevant. Finally, 9 original studies fulfilled the inclusion criteria and were further scrutinized [30,31,34,35,36,37,38,39,40]. The PRISMA flow diagram of the literature search and study selection is shown in Figure 3.

All 9 studies were retrospective observational studies, mostly published after 2015. The majority were conducted in German and Dutch institutions. Their data are summarized in Table 1. Patients in the included studies were subjected to either McKeown or Ivor Lewis esophagectomy with GC reconstruction, except for 1 patient in the study of Schröder et al., in whom a colon interposition was performed [34]. Jefferies et al. did not report the type of esophagogastric anastomosis [40]. The AL rate ranged from 8.5 to 24% among studies.

### 3.1. Association of Arterial Calcification with Anastomotic Leakage after Esophagectomy

Van Rossum et al. were the first to propose a visual scoring system for arterial calcification on preoperative CT [35]. This scoring system is based on the assessment of the presence and degree of calcification of selected arteries, and is evaluated on axial CT images on a slice-by-slice basis. The authors retrospectively evaluated the preoperative CT scans of 246 patients operated with McKeown esophagectomy, and visually calculated the calcification score on the thoracic and upper abdominal aorta, celiac axis, right post-celiac arteries (common hepatic artery, gastroduodenal artery, and right gastroepiploic artery), and left post-celiac arteries (splenic artery and left gastroepiploic artery). Multivariate regression analysis of the individual scores revealed an association between AL and minor (OR, 2.00; 95% CI: 1.02, 3.94), as well as major (OR, 2.87; 95% CI: 1.22, 6.72), aortic calcifications. These findings were confirmed in 167 patients treated with Ivor Lewis esophagectomy by Goense et al., using the same scoring system [37]. Borggreve et al. [39] evaluated the visual scoring system developed by van Rossum [35] in additional trajectories. Calcification in the coronary arteries, supra-aortic arteries, and thoracic aorta was associated with AL, which indicates that generalized cerebrovascular and cardiovascular disease might be a risk factor for AL. Conversely, the study failed to demonstrate a significant association between calcification in the abdominal aorta or the celiac trunk/right post-celiac arteries and AL.

Zhao et al. [36] included 673 patients after McKeown esophagectomy in a Chinese population. Instead of grading the degree of arterial calcifications, this group used a simple binary scoring system based on the presence or absence of calcifications and reported a significantly higher rate of AL in patients with calcification of the aorta, celiac trunk, and right/left post-celiac arteries. In contrast to the aforementioned studies, Jefferies et al. [40] could not demonstrate any statistically significant association between arterial calcification and AL after performing a retrospective analysis of preoperative CT scans from 411 patients who underwent esophagogastric anastomosis.

### 3.2. Association of Celiac Trunk Stenosis with Anastomotic Leakage after Esophagectomy

Schröder et al. (2002) laid the groundwork for investigating the relationship between celiac trunk stenosis and AL [34]. The group quantified the degree of celiac trunk stenosis by conventional mesenterico-celiacography in 23 patients undergoing esophagectomy, but failed to demonstrate statistically significant differences [34]. Lainas et al. [31] assessed the preoperative arterial phase CT scans of 481 patients undergoing Ivor Lewis esophagectomy, and classified the celiac trunk as normal, with extrinsic stenosis due to a median arcuate ligament, or with intrinsic stenosis due to atherosclerotic changes [31]. The study revealed higher rates of gastric conduit necrosis in patients with intrinsic or extrinsic stenosis compared to those with a normal celiac trunk. However, the study did not assess the degree of celiac trunk stenosis comprehensively, nor was multivariate analysis performed to evaluate whether celiac trunk stenosis is an independent risk factor for AL.

Two other studies conducted at the same institution between January 2014 and December 2014 included 164 and 154 patients who had undergone Ivor Lewis esophagectomy, respectively [30,38]. In both studies, the NASCET formula was used to assess the degree of celiac trunk stenosis on preoperative CT, and was successfully correlated with AL after esophagectomy. In addition, Chang et al. [38] performed a visual assessment of calcium with the scoring system proposed by van Rossum et al. [35] on the same patient group. No statistically significant association between arterial calcification and AL was found, suggesting that the assessment of celiac trunk stenosis might be a more robust method to predict AL.

## 4. Discussion

The majority of the studies included in this review indicate that arterial calcification and celiac trunk stenosis are associated with an increased risk for AL after esophagectomy with GC reconstruction. However, the presence of studies that do not reproduce the aforementioned association generates questions to be answered and issues for further assessment.

This up-to-date literature review adds value to the previously published reviews by Knight et al. [28] and Hoek et al. [29] concerning the impact of arterial calcification in AL after esophagectomy, by including studies evaluating another radiological risk factor for AL, namely celiac trunk stenosis. Given the fact that the arterial supply of the reconstructed GC which will be anastomosed to the esophageal remnant derives invariably from the right gastroepiploic artery [25], a branch of the celiac trunk, it is rational to assume that stenosis in the celiac trunk may lead to increased rates of GC necrosis and anastomotic leakage following esophagectomy. This hypothesis could not be initially verified by Schröder et al., maybe due to the inclusion of a small number of patients and events (5 out of 23 patients developed AL) [34]. Furthermore, the method utilized for the assessment of stenosis, namely mesenterico-celiacography, is an outdated method which is replaced by CT angiography nowadays [41,42]. Later studies with larger cohorts and an assessment of stenosis on arterial phase CT, based both on a visual evaluation [31] and the calculation of the degree of stenosis with the NASCET formula [30,38], confirmed the association between celiac trunk stenosis and AL after esophagectomy. Automated and semi-automated segmentation methods have long been established for the assessment of stenosis in carotid arteries on arterial phase CT, and seem to generate more robust measurements compared to manual methods [43]. Thus, the same techniques could be applied to quantify celiac trunk stenosis, automating the evaluation process of this risk factor for AL post esophagectomy. From the included reviewed original articles, cut-off values concerning the different degrees of celiac trunk stenosis associated with AL was not feasible, due to a varying individual collateral blood flow via the superior mesenteric artery and a wide spectrum of anatomical variations in the stomach’s greater curvature arterial blood supply [30,31,38]. Therefore, future studies need to examine not only the anatomical vascular variations, but also the functional vascular changes, by measuring the arterial blood flow that the gastric conduit receives [30].

Regarding the relation of arterial calcification with AL following esophagectomy, not all the studies included succeeded in identifying a statistically significant association. One possible reason for the large variation in reported results might be the use of a visual calcification score that can be radiologist-dependent and, additionally, susceptible to miscalculations due to anatomical and physiological variations among patients. Moreover, the study cohorts were heterogenic, including different esophageal cancer types, and variable percentages of given neoadjuvant therapies, as well as different surgical techniques. The aforementioned reasons are also possible factors in the wide variation in the reported incidence of AL. Due to the discrepancies in the above-mentioned results, Hoek et al. [29] conducted a meta-analysis pooling data from all studies investigating the relation between arterial calcification and AL after esophagectomy with GC reconstruction [29]. The meta-analysis showed a significant association between high calcium score and AL for the thoracic aorta, celiac trunk, and right post-celiac axis, confirming the hypothesis that atherosclerotic disease can lead to AL. One possible way to reduce variation in the assessment of arterial calcification is to introduce a scoring system based on a quantitative, rather than visual, evaluation of the calcification burden. Such a scoring system has been successfully applied to coronary arteries for the assessment of cardiovascular risk [44], and could be easily implemented for other arteries, such as the thoracic aorta.

Identifying patients at risk for anastomotic leakage after esophagectomy could significantly alter patient management, and subsequently decrease the postoperative morbidity and mortality [45]. Preoperative strategies for reducing the incidence of AL post esophagectomy could include adequate patient selection and pre-habilitation, as well as preconditioning of the stomach. The latter is an innovative technique that involves the preoperative coiling or surgical occlusion of all supplying vessels to the stomach except the right gastroepiploic artery, to increase the vascularization of the future GC [46,47]. Lammerts et al. [33] demonstrated that percutaneous angioplasty is feasible in patients with significant celiac trunk stenosis prior to Ivor Lewis esophagectomy. Perioperatively, high-risk patients could be managed with patient-specific surgical and anesthesiologic strategies, taking into consideration the choice of surgical procedure and anastomotic technique [48], as well as intraoperative perfusion monitoring and postoperative fluid and inotrope management [45,48,49,50]. After surgery, GC decompression using naso-gastric tube, early diagnostic endoscopy before initiation of oral nutrition, and pre-emptive vacuum therapy could be employed both for early detection of possible GC necrosis and improvement of the arterial perfusion of the reconstructed GC to reduce the risk for AL and its consequences [51,52].

Obviously, there are several other factors besides blood flow that are important for AL. Potential risk factors for AL after esophagectomy, such as age, sex, high body mass index, hyperlipidemia, malnutrition, smoking, hypertension, hypotension, prior neoadjuvant therapy, duration of surgery, hospital volume, and chronic use of steroids, American Society of Anesthesiologists (ASA) score, chronic obstructive pulmonary disease (COPD), forced expiratory volume in one second (FEV1), diffusion capacity for carbon monoxide of the lung (DLCO), and other comorbidities (e.g., coronary artery disease, cardiovascular disease, diabetes mellitus and renal insufficiency) were tested in the included reviewed studies, as variables in logistic regression models to evaluate whether those parameters could be independently associated with AL [30,31,34,35,36,37,38,39,40]. The large variety in the aforementioned parameters and the different statistical approaches of each study result in conflicting conclusions. Multivariable logistic regression models in the studies by van Rossum et al. [35], Goense et al. [37], Borggreve et al. [39], and Jefferies et al. [40] did not reveal any independent association of the above-mentioned risk factors with AL, revealing the multifactorial nature of this complication after esophagectomy. On the other hand, using a multivariable logistic regression, the study by Zhao et al. [36] revealed a significant association (*p* < 0.05) of AL with demographic and clinical characteristics of patients such as ASA score, prior thoracic surgery, upper digestive tract ulcer, COPD, hypertension, peripheral vascular disease, renal insufficiency, FEV1, and DLCO. In a univariable analysis by Chang et al. [38], no significant difference in tumor type and other preoperative comorbidities was found. However, a univariable analysis by Brinkmann et al. [30] showed a higher rate of AL in patients with squamous cell carcinoma, but no significant differences concerning preexisting comorbidities among patients with or without AL. Of note is that in the study of Lainas et al. [31], 319 patients underwent neoadjuvant therapy, but no significant differences were found between those who developed conduit necrosis and those without conduit necrosis (*p* = 0.732).

There are limitations to a review like this. First, the vast majority of the included articles are single-institution, retrospective studies and therefore subject to selection and reporting bias. Second, as already mentioned, the cohort groups are heterogenic, and the outcomes could be difficult to interpret and compare. On the other hand, one strength of the current review is that all available published evidence in the field is included, and the key message is that well-designed randomized controlled trials or large population-based cohort studies are warranted.

## 5. Conclusions

In conclusion, arterial calcification and celiac trunk stenosis seem to be reliable risk factors for predicting anastomotic leakage after esophagectomy with GC reconstruction, and allow the modification of management of high-risk patients to reduce postoperative complications. Both risk factors can be easily assessed on widely available preoperative CT scans. Future studies should evaluate the use of automated quantitative methods for the assessment of arterial calcification and celiac trunk stenosis. In that way, the reported results could be not only directly comparable but also reproducible.

## Figures and Tables

**Figure 1 cancers-14-01016-f001:**
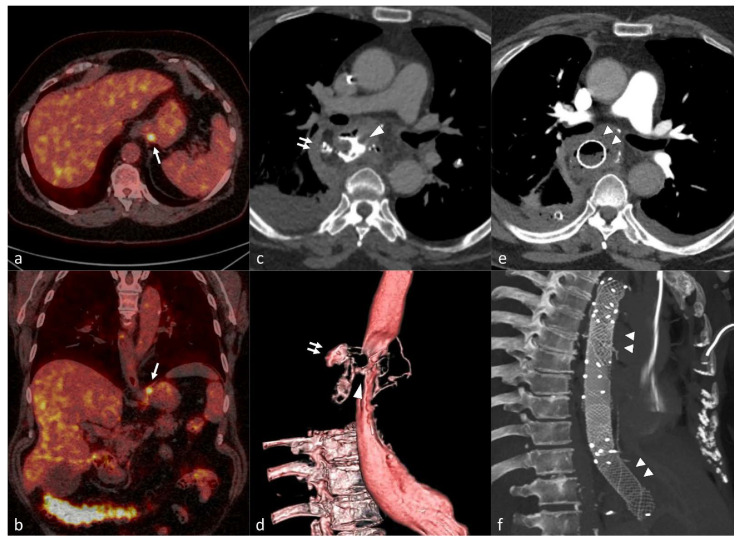
Preoperative axial (**a**) and oblique coronal (**b**) fused images of ^18^F-FDG PET/CT * showing focal FDG uptake in the gastric cardia (single arrow) in a 63-year-old female patient with gastric cardia adenocarcinoma. The patient underwent minimally invasive esophagectomy with gastric tube reconstruction (Ivor Lewis) after neoadjuvant chemotherapy. Axial (**c**) and 3D reconstruction (**d**) images of postoperative CT scan with oral contrast on the 6th postoperative revealing leakage (double arrows) due to a large defect in the gastric conduit wall (arrowhead). Three overlapping stents were endoscopically placed to cover the defect as seen on the axial (**e**) and oblique sagittal MIP ^†^ (f) of the follow-up CT scan (double arrowheads). * ^18^F-FDG PET/CT: 18F-Flurodeoxyflucose Positron Emission Tomography/Computed Tomography. ^†^ MIP: Maximum Intensity Projection.

**Figure 2 cancers-14-01016-f002:**
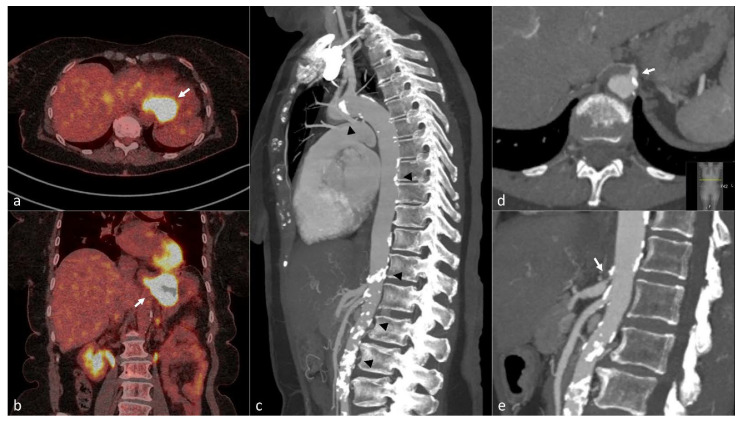
A 66-year-old woman with squamous cell carcinoma of the gastric cardia. Fused axial (**a**) and coronal (**b**) image of preoperative ^18^F-FDG PET/CT * scan shows the tumor in distal esophagus and gastric cardia with strong FDG uptake (white arrows). Sagittal MIP ^†^ (**c**) of arterial phase Computed Tomography revealed a high burden of atherosclerosis with multiple large, calcified plaques in the thoracic and abdominal aorta (black arrow heads). Enlarged axial (**d**) and sagittal (**e**) images of the same examination shows a calcified plaque causing significant stenosis in the celiac axis (double arrows). The patient underwent minimally invasive esophagectomy (Ivor-Lewis) after chemoradiotherapy and developed anastomotic leakage on the 8th postoperative day. * ^18^F-FDG PET/CT: 18F-Flurodeoxyflucose Positron Emission Tomography/Computed Tomography. ^†^ MIP: Maximum Intensity Projection.

**Figure 3 cancers-14-01016-f003:**
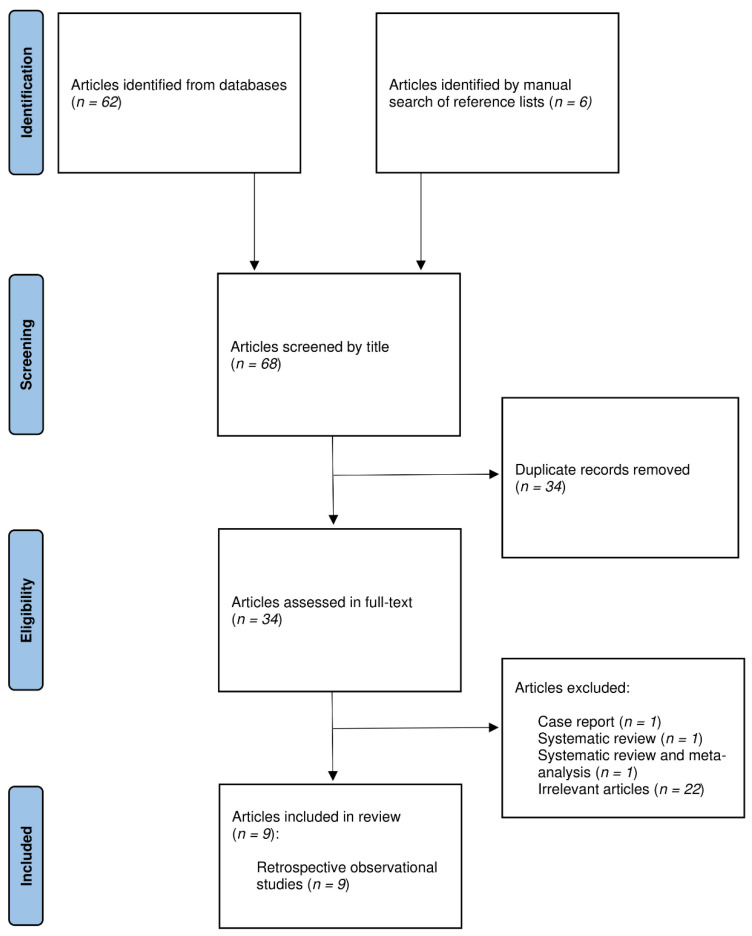
PRISMA flow chart.

**Table 1 cancers-14-01016-t001:** Summary data of included studies investigating the association of arterial calcification and/or celiac trunk stenosis with anastomotic leakage after esophagectomy with gastric conduit reconstruction.

Author/Country/ Year	Type of Study (Time Period)	Type of Surgery	Surgical Approach (Open/MIE ^†^/Hybrid)	Neoadjuvant Treatment (Total Number of Patients)	AL ^§^ Rate (%)	Association of Arterial Calcification with AL (Trajectories)	Association of Celiac Trunk Stenosis with AL	Findings	Significance
Schröder/Germany/2002 [34]	Prospective (NS) *	Ivor Lewis (*n* = 15), McKeown (*n* = 7) and colon interposition (*n* = 1)	Open	14 (23)	21	n/a	-	Celiac trunk stenosis was not associated with AL.	First study that investigated correlation of celiac artery stenosis and AL.
van Rossum/The Netherlands/2015 [35]	Retrospective (2003–2012)	McKeown (*n* = 168) and transhiatal (*n* = 78)	Open (*n* = 42) MIE (*n* = 190) Hybrid (*n* = 14)	134 (246)	24	+ ^±^ (aorta, right post-celiac arteries)	n/a	Calcifications of the aorta and the right post-celiac arteries were independently associated with AL.	First study that proposed a visual scoring system for arterial calcification, and demonstrated association of aortic calcification with AL.
Zhao/China/2016 [36]	Retrospective (2010–2015)	McKeown	Open (*n* = 264) MIE (*n* = 348) Hybrid (*n* = 97)	80 (709)	17.2	+ (aorta, celiac axis)	n/a	Calcifications of the aorta and the celiac axis were independently associated with AL.	Demonstrated that presence of calcification in the aorta or celiac artery are independent risk factors for AL in a Chinese population.
Goense/The Netherlands/2016 [37]	Retrospective (2012–2015)	Ivor Lewis	MIE	153 (167)	24	+ (aorta)	n/a	Calcifications of the aorta was independently associated with AL, while calcification of the celiac axis, left and right post-celiac arteries were not.	Demonstrated that presence of calcification on the aorta is an independent risk factor for AL. No significant association for calcification of other arteries.
Lainas/France/2017 [31]	Retrospective (2004–2014)	Ivor Lewis	Open (*n* = 239) Hybrid (*n* = 242)	319 (481)	17.4	n/a	+	Celiac trunk stenosis was independently associated with gastric conduit necrosis. Also, AL occurred more often in patients with celiac trunk stenosis.	Investigated the correlation of celiac trunk stenosis (including extrinsic, caused by median arcuate ligament compression, and intrinsic, caused by calcifications) and gastric conduit necrosis.
Chang/Germany/ 2018 [38]	Retrospective (2014)	Ivor Lewis	Open and hybrid	n/a ^‡^ (164)	8.5	- ^#^	+	Celiac trunk stenosis was associated with AL, while calcifications in the aorta, celiac axis, the left and right post-celiac arteries were not.	Found association of celiac trunk stenosis with AL, but no association of arterial calcifications and AL.
Borggreve/ The Netherlands/2018 [39]	Retrospective (2003–2015)	McKeown (*n* = 308) and transhiatal (*n* = 98)	Open (*n* = 80) MIE (*n* = 311) Hybrid (*n* = 15)	275 (406)	25.6	+ (supra-aortic arteries, coronary arteries)	n/a	Calcifications of the supra-aortic arteries and the coronary arteries were independently associated with AL. No significant association was found between the calcifications of the celiac axis or abdominal aorta, and AL.	Suggests that generalized cerebrovascular disease is a strong indicator for risk of AL.
Jefferies/United Kingdom/2019 [40]	Retrospective 2006–2018)	Ivor Lewis (*n* = 379) and McKeown (*n* = 34)	Open (*n* = 86) MIE (*n* = 103) Hybrid (*n* = 224)	344 (413)	15.8	-	n/a	The presence of calcification at several sites including the celiac axis, post-celiac arteries, the proximal and distal aorta was studied, and no association with AL or gastric conduit necrosis was found.	No significant association between arterial calcification and AL or gastric conduit necrosis.
Brinkmann/ Germany/2019 [30]	Prospective (2014)	Ivor Lewis	Open (*n* = 17) Hybrid (*n* = 137)	124 (154)	9.7	n/a	+	Celiac trunk stenosis was independently associated with AL.	Demonstrated that celiac trunk stenosis is an independent risk factor for AL.

* NS, not specified. ^†^ MIE, minimally invasive esophagectomy. ^‡^ n/a, non-applicable/not available. ^§^ AL, anastomotic leakage. ^±^ +, association was found. ^#^ -, no association was found.

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
