# Peer review of "Role of Radiology in the Preoperative Detection of Arterial Calcification and Celiac Trunk Stenosis and Its Association with Anastomotic Leakage Post Esophagectomy, an Up-to-Date Review of the Literature"

_cancers, 2022, doi:10.3390/cancers14041016_

Round 1
Reviewer 1 Report
This review aims to study the current literature regarding the role of radiology in detecting potential risk factors associated with anastomotic leakage.
Hoek et al has already published a systematic review and meta-analysis about the same theme in 2020. The authors mentioned that
this up-to-date literature review adds value to the previously published reviews by Knight et al. and Hoek et al. concerning the impact of arterial calcification in AL after esophagectomy by including studies evaluating another radiological risk factor for AL, namely celiac trunk stenosis.
However, it seems that there is any new value in this paper.
Author Response
Reviewer #1: This review aims to study the current literature regarding the role of radiology in detecting potential risk factors associated with anastomotic leakage.
Hoek et al has already published a systematic review and meta-analysis about the same theme in 2020. The authors mentioned that this up-to-date literature review adds value to the previously published reviews by Knight et al. and Hoek et al. concerning the impact of arterial calcification in AL after esophagectomy by including studies evaluating another radiological risk factor for AL, namely celiac trunk stenosis.
However, it seems that there is any new value in this paper.
A. We agree with the reviewer that a systematic review and meta-analysis on similar topic has been recently published, which is also mentioned in the current review.
However, in the previously published review by Knight et al and the very well written metanalysis by Hoek et al the effect of the celiac trunk stenosis as a risk factor for anastomotic leakage (AL) is not extensively discussed. In our review we focused on the preoperative radiological assessment emphasizing both in the grade of celiac trunk stenosis and the calcification burden of the aorta as risk factors for AL. Additionally, we included two studies, by Lainas et al and by Schröder et al, that were absent from the aforementioned review/metanalysis. The study by Lainas et al included 481 patients showing that celiac trunk stenosis was independently associated with gastric conduit necrosis. The study by Schröder et al was the first study that investigated correlation of celiac trunk stenosis and AL. The group quantified the degree of celiac trunk stenosis by conventional mesenterico-celiacography in 23 patients undergoing esophagectomy but failed to demonstrate statistically significant differences.

Reviewer 2 Report
This is a review article on the relationship between arterial calcification and celiac trunk stenosis and anastomotic leakage post esophagectomy, but I think it would be better to have a little deeper content and explanation. I have some comments to the author.
Comment 1. I think that the effect of blood flow is important for anastomotic leakage, but I think that other factors such as nutritional status, serum albumin level and background disease (ex. presence or absence of DM) are also related, but this systematic How did the cited references that were the subject of the review deal with this background? Also, Table 1 has a column for neoadjuvant treatment, but did neoadjuvant treatment have any effect on anastomotic leakage?
Comment 2. Arterial calcification and celiac trunk stenosis are variously defined in the cited references (although some are not clearly defined), but the definitions and degrees of these definitions are summarized in a little more detail, and in fact Why not give a little more consideration about what kind of definition is desirable for calcification and stenosis in order to consider the arterial blood flow of?
Author Response
Reviewer #2: This is a review article on the relationship between arterial calcification and celiac trunk stenosis and anastomotic leakage post esophagectomy, but I think it would be better to have a little deeper content and explanation. I have some comments to the author.
Comment 1. I think that the effect of blood flow is important for anastomotic leakage, but I think that other factors such as nutritional status, serum albumin level and background disease (ex. presence or absence of DM) are also related, but this systematic How did the cited references that were the subject of the review deal with this background? Also, Table 1 has a column for neoadjuvant treatment, but did neoadjuvant treatment have any effect on anastomotic leakage?
A. We totally agree with the reviewer that several other factors, besides blood flow, are important for AL. Since the current review focused on the preoperative radiological assessment of aortic calcification and celiac trunk stenosis as risk factors for AL after esophagectomy we chose not to analyse in depth other risk factors contributing to AL. Those factors are mentioned briefly in the introduction accompanied by the appropriate reference. Moreover, in the revised version of the manuscript we have now added an extra paragraph in the ¨Discussion¨ which reviews other risk factors for AL as assessed in the studies included in our reference list. With regards to neoadjuvant treatment as risk factor for AL was specifically addressed in the study by Lainas et al. In this study, 319 patients underwent neoadjuvant therapy prior to esophagectomy but no significant differences found between those who developed conduit necrosis and those without conduit necrosis (P=0.732).
Comment 2. Arterial calcification and celiac trunk stenosis are variously defined in the cited references (although some are not clearly defined), but the definitions and degrees of these definitions are summarized in a little more detail, and in fact Why not give a little more consideration about what kind of definition is desirable for calcification and stenosis in order to consider the arterial blood flow of?
A. This is a valuable comment that should be studied/evaluated in future original studies. As far as it concerns the calcification burden of the aorta it has been stated how the different studies evaluated it (e.g., by using the calcium score proposed by van Rossum et al or by using a binary scoring system depending on the presence or absence of aorta calcifications proposed by Zhao et al). Concerning the definition of celiac trunk stenosis (due to extrinsic or intrinsic/atherosclerotic factors), cut off value for the degree of celiac trunk stenosis associated with AL was not feasible to be calculated due to an individual collateral blood flow via superior mesenteric artery and a wide anatomical variety of stomach`s greater curvature blood supply. A clarifying comment on the latter has been now added in the Discussion¨ of the revised manuscript.

Reviewer 3 Report
I read with interest the paper submitted by A Tzortzakakis et al.
Minor comments: the arrows in figure 2 are not visible.
I don’t understand the meaning of the 2 questions at the endo of the paragraph “introduction”., since these questions are only partially discussed in the paragraph “discussion” but are not part of the results of the paper. They can be omitted or changed.
Author Response
Reviewer #3: I read with interest the paper submitted by A Tzortzakakis et al.
Minor comments: the arrows in figure 2 are not visible.
A. The arrows/arrowheads have now been edited so that they can be more visible. We have reuploaded figures 1-2 and corresponding figure legends with bigger arrows/arrowheads.
I don’t understand the meaning of the 2 questions at the endo of the paragraph “introduction”., since these questions are only partially discussed in the paragraph “discussion” but are not part of the results of the paper. They can be omitted or changed.
A. The questions in the last paragraph of the introduction have now been omitted and the paragraph has been rephrased accordingly.
